# CabbageNet: Deep Learning for High-Precision Cabbage Segmentation in Complex Settings for Autonomous Harvesting Robotics

**DOI:** 10.3390/s24248115

**Published:** 2024-12-19

**Authors:** Yongqiang Tian, Xinyu Cao, Taihong Zhang, Huarui Wu, Chunjiang Zhao, Yunjie Zhao

**Affiliations:** 1School of Computer and Information Engineering, Xinjiang Agricultural University, Urumqi 830052, China; 2Ministry of Education Engineering Research Center for Intelligent Agriculture, Urumqi 830052, China; 3Xinjiang Agricultural Informatization Engineering Technology Research Center, Urumqi 830052, China; 4National Engineering Research Center for Information Technology in Agriculture, Beijing 100125, China; 5Key Laboratory of Digital Village Technology, Ministry of Agriculture and Rural Affairs, Beijing 100125, China

**Keywords:** cabbage, deep learning, instance segmentation, automatic harvesting, intelligent agriculture

## Abstract

Reducing damage and missed harvest rates is essential for improving efficiency in unmanned cabbage harvesting. Accurate real-time segmentation of cabbage heads can significantly alleviate these issues and enhance overall harvesting performance. However, the complexity of the growing environment and the morphological variability of field-grown cabbage present major challenges to achieving precise segmentation. This study proposes an improved YOLOv8n-seg network to address these challenges effectively. Key improvements include modifying the baseline model’s final C2f module and integrating deformable attention with dynamic sampling points to enhance segmentation performance. Additionally, an ADown module minimizes detail loss from excessive downsampling by using depthwise separable convolutions to reduce parameter count and computational load. To improve the detection of small cabbage heads, a Small Object Enhance Pyramid based on the PAFPN architecture is introduced, significantly boosting performance for small targets. The experimental results show that the proposed model achieves a Mask Precision of 92.2%, Mask Recall of 87.2%, and Mask mAP50 of 95.1%, while maintaining a compact model size of only 6.46 MB. These metrics indicate superior accuracy and efficiency over mainstream instance segmentation models, facilitating real-time, precise cabbage harvesting in complex environments.

## 1. Introduction

Cabbage, as one of the extensively cultivated vegetable crops in China, leads globally in terms of planting area and output, reaching approximately 900,000 hectares and 35,000,000 tons [1]. The complexity of the environmental conditions faced during cabbage harvesting significantly inhibits the efficiency and precision of mechanical harvesting, thereby rendering the process highly dependent on manual labor and physical resources [2,3]. The intimate fusion of artificial intelligence technology with agricultural practices has gradually emerged as a critical trend in the evolution of contemporary agriculture, positioning smart agriculture at the forefront of this advancement [4]. Particularly for cabbage harvesting, accurate recognition and segmentation of cabbage heads are the cornerstone technologies for achieving automated harvesting. This technology promises to facilitate unmanned and automated cabbage harvesting processes, reduce production costs and labor intensity, while simultaneously enhancing the precision and efficiency of harvesting [5]. Moreover, it assists in the non-destructive prediction of cabbage yield. Consequently, the study of technologies for the identification and segmentation of cabbage heads is of paramount significance.

Traditional image segmentation techniques rely on threshold selection, edge detection, and region growth to divide the image, making them more accurate for the segmentation of simple scenes. However, achieving segmentation accuracy for complex scenes remains a challenge [6]. The development of machine vision technology has led to the maturation of image segmentation technology based on deep learning, with an increasing number of image segmentation algorithms being proposed [7]. In contrast to traditional image segmentation algorithms, deep learning-based algorithms are capable of capturing more complex and detailed texture characteristics of the target object through the training of neural network models, thereby markedly enhancing the efficacy of segmentation tasks in a variety of complex environments [8].

Deep learning-based image segmentation techniques, such as Mask R-CNN [9], YOLACT [10], and ConvNeXt V2 [11], have demonstrated strong performance in agricultural robotics, particularly in crop, vegetable, and fruit detection for automated harvesting [12]. Shuo Kang et al. [13] employed DeepLabV3+ for broccoli head detection, achieving 57.9% mIoU and 98.56% pixel accuracy, but their model requires 0.75 s per image, limiting real-time applicability. Similarly, Pieter M. Blok et al. [14,15] employed an Occlusion Region-based Convolutional Neural Network (ORCNN) for broccoli segmentation, achieving a 6.4 mm dimensional discrepancy, but its slow processing prevents real-time deployment in field environments. Hanwen Kang et al. [16] proposed a Geometry-Aware (A3N) network for fruit identification in orchards, achieving an 87.3% instance segmentation accuracy in 35 ms. However, its low accuracy and focus on orchard environments limit its use in more complex settings. Lei Shen et al. [17] integrated attention mechanisms with Mask R-CNN for grape cluster segmentation, achieving 59.1% accuracy, but the model’s low frame rate and accuracy hinder real-world, high-speed applications. Similarly, Dandan Wang et al. [18] proposed an enhanced Mask R-CNN for apple segmentation, with 93.2% accuracy for bounding box detection, but its 0.27 s/image (3.7 fps) processing time is insufficient for real-time tasks. Gabriel Coll-Ribes et al. [19] used CNN-based instance segmentation and monocular depth estimation for grape bunch and peduncle segmentation. Despite effectiveness, its 0.5 FPS frame rate is inadequate for real-time dynamic environments. Olarewaju Mubashiru Lawal [20] developed YOLOv5n-based YOLOv5-LiNet for cucurbit fruit segmentation, achieving 88.5% accuracy with real-time capabilities, but it struggles with occlusion, limiting its performance in complex scenarios. Yajun Li et al. [21] proposed an MTA-YOLACT-based algorithm for tomato cluster segmentation, achieving high accuracy, but its demand for high agronomic precision limits its applicability in open-field cabbage harvesting. Nils Lüling et al. [22,23,24] designed a Mask R-CNN algorithm for cabbage volume and leaf area assessment, achieving 92.6% and 89.8% accuracy, but its processing speed hinders real-time use. Masaki Asano et al. [25] employed an SSD-based approach for autonomous cabbage harvesting, but suffers from low recognition accuracy. Peichao Cong et al. [26] integrated Swin Transformer attention mechanisms into Mask R-CNN for sweet pepper segmentation, but its speed is inadequate for deployment in open-field agricultural settings that require real-time processing with limited computational resources. Huarui Wu et al. [27] proposed an UperNet model with a Swin Transformer backbone for Brassica napus segmentation, achieving 91.2% mIoU and 95.2% pixel accuracy, but its processing speed is too slow for use in real-time harvesting systems that require quick and efficient decision-making. Jia Weikuan et al. [28] enhanced SOLO with ResNeSt and FPN, achieving 94.84% recall and 96.16% precision for persimmon and apple segmentation, but its computational demands make it difficult to implement in real-time agricultural scenarios. Xing Sheng et al. [29] proposed EdgeSegNet for fruit segmentation in complex scenes, achieving 90.9% and 94.2% mIoU for apple and peach, but its processing speed limits real-time performance.

The studies mentioned above have improved upon deep learning-based segmentation algorithms to achieve fruit and leaf segmentation. However, accurately segmenting fruits and vegetables in complex backgrounds, especially when they are in motion, remains a significant challenge. In particular, the segmentation of cabbage heads during the harvesting process is further complicated by the need for high accuracy, real-time performance, and computational efficiency. Achieving precise, efficient, and lightweight segmentation in dynamic and cluttered field environments requires models that balance segmentation accuracy with processing speed to meet the demands of automated harvesting. To address these challenges, this study proposes an enhanced YOLOv8n [30] instance segmentation model, “CabbageNet”. Designed to overcome issues such as inaccurate detection of moving cabbage heads, inefficiencies, and misidentification or missed detection of small cabbages, CabbageNet ensures precise and efficient segmentation of cabbage heads in complex harvesting scenes. The main contributions of this study are as follows:Construction of a cabbage instance segmentation dataset: Cabbage images from the harvest period were collected using image acquisition devices and search engines. After rigorous selection and processing, 10,000 images were annotated, creating a high-quality cabbage head instance segmentation dataset.Integration of deformable attention into the C2f module: The C2f module was enhanced through the incorporation of deformable attention and dynamic sampling points, forming a deformable attention module. This enables the model to better adapt to varying image sizes and content, thereby improving accuracy and efficiency in instance segmentation tasks.Improvement of the downsampling process using the ADown module: The ADown module employs an adaptive mechanism to retain essential information and capture higher-level image features. The model efficiently handles objects of varying sizes through multi-scale feature fusion, improving both accuracy and robustness in instance segmentation tasks.Enhancement of small object segmentation using the SOEP module: The Small Object Enhance Pyramid (SOEP) applies Spatial-Depthwise Separable Convolution (SPDConv) to the P2 layer to extract richer small-object features. Combined with CSP-OmniKernel feature aggregation, SOEP effectively preserves critical small-object information, significantly improving segmentation performance for small objects.

## 2. Materials and Methods

### 2.1. Image Acquisition

In this study, the collection process of cabbage head image data was divided into two main parts. The first part includes 6000 images, all collected on site. The collection sites included Jintaiyang Farm in Changping District, Beijing, the National Precision Agriculture Research Demonstration Base in Xiaotangshan, and the experimental base in Canal District, Cangzhou, Hebei Province. The collection focused on the Zhonggan-21 cabbage variety, a widely cultivated species in North China. To ensure the diversity and comprehensiveness of the dataset, images were selected under various climatic conditions, illumination levels, shooting angles, and cabbage growth stages. The resolution of the image collection was set to 3024 × 3024 to ensure that the image quality met the study’s specifications. The second part included 4000 images obtained from the Internet through web crawling techniques, thereby expanding the breadth and diversity of the data. Throughout the data collection process, keywords related to cabbage heads were first identified, followed by thorough searches on the Internet using these keywords. This approach ensured a rich quantity of images. Following the acquisition of over 20,000 original images, a series of stringent screening procedures was undertaken to guarantee the content, clarity, and relevance of the images. Through this meticulous process, the scientific relevance and research value of the images were greatly enhanced. Ultimately, 4000 images were carefully selected from the initial pool, adhering to the study’s requirements.

Throughout the entire data collection process, we endeavored to encompass all scenarios encountered during the actual harvest, ensuring the comprehensiveness and breadth of the data. Following a meticulous preliminary phase of data organization and rigorous screening, we successfully compiled a collection of 10,000 cabbage heads images. Each collected image strictly meets the standards for dataset construction in scientific research.

### 2.2. Dataset Establishment

The construction of datasets is crucially important for training the models. Initially, the filtered images underwent a uniform resizing process, standardizing all images to 640 × 640 pixels as required. Subsequently, labeling was conducted for instance segmentation using X-AnyLabeling (v2.3.5, by Wei Wang, CVHub organization, published on GitHub) [31] for annotation purposes. Data marking before and after labeling is illustrated in Figure 1. Then, the annotated JSON files were converted into both YOLO and COCO dataset formats. The YOLO-format dataset was used to train the model proposed in this paper, while the COCO format was utilized for comparative experiments with other models. The transformed datasets were then randomly partitioned into training, validation, and testing sets with a ratio of 6:2:2. The training set was used to train the model to fit the data distribution pattern, involving the determination of model parameters such as weights and biases. The validation set was used to adjust model parameters and hyperparameters to optimize performance and prevent overfitting. Finally, the testing set was used to evaluate the generalization capability of the model.

### 2.3. Network Structure

The YOLO (You Only Look Once) algorithm is a one-stage object detection method. In YOLOv8, the algorithm has been further extended with a variant specifically designed for segmentation, known as YOLOv8-seg. This variant is optimized for segmentation tasks by simultaneously predicting bounding boxes, class probabilities, and pixel-level masks. In YOLOv8-seg, the architecture includes two distinct heads: a Detection Head for bounding box coordinates and class probabilities, and a Segmentation Head for generating pixel-level masks. In this paper, the necessity of timeliness in unmanned cabbage harvesting is investigated, and CabbageNet is proposed by choosing YOLOv8n-seg as the baseline model, which enhances segmentation accuracy and efficiency for targets of various sizes by integrating a deformable attention mechanism in the C2f module and introducing the C2f deformable module. Multi-scale feature fusion is achieved by enhancing the ADown module, thereby improving segmentation accuracy for various sizes of targets and ensuring robust performance in the instance segmentation task. Ultimately, the SOEP module enhances the segmentation performance of small objects, facilitating the effective segmentation of cabbage heads during harvesting in complex backgrounds. Figure 2 illustrates the structural diagram of the CabbageNet model network, detailing its architecture and components.

#### 2.3.1. C2f Deformable Attention Block

In this paper, the final C2f module in the BottleNeck architecture is enhanced by incorporating a deformable attention module [32,33], as shown in Figure 3, introducing a deformable attention mechanism and dynamic sampling points to improve model performance. The traditional Transformer uses standard self-attention, requiring the processing of all pixels in the image, which significantly increases computation. To address this challenge, in this paper, we choose the deformable attention mechanism, which focuses on key regions of the image, reducing computational load while maintaining performance. Furthermore, the deformable attention mechanism dynamically selects sampling points, allowing the model to better focus on critical regions. By incorporating the deformable attention module, the model adapts better to images of varying sizes and contents, improving efficiency and accuracy in instance segmentation tasks.

#### 2.3.2. ADown Block

The ADown module [34] is a convolutional structure that has been specifically designed for the purpose of downsampling operations in deep learning models, as illustrated in Figure 4. In such models, downsampling serves as a crucial technique for the reduction in the spatial dimensions of feature maps. This reduction allows the model to capture image features at higher levels while simultaneously reducing the computational costs related to that process. The ADown module employs an adaptive mechanism that determines whether to transmit deeper features based on the variability of information across different regions of the input feature map, facilitating a flexible downsampling strategy that effectively preserves critical information. Moreover, the ADown module utilizes a multi-scale feature fusion approach that combines features from various scales, thereby enabling the adept processing of targets of different sizes. This mitigates the detailed loss associated with excessive downsampling and enhances detection accuracy. It is noteworthy that the incorporation of depthwise separable convolutions allows the ADown module to significantly reduce the number of parameters and computational complexity while maintaining a high level of feature extraction capability. As a result, the ADown module markedly improves the model’s accuracy and computational efficiency, ensuring robust performance in instance segmentation tasks even within complex backgrounds.

#### 2.3.3. Small Object Enhance Pyramid

In instance segmentation tasks with YOLOv8, the standard P3, P4, and P5 feature layers often encounter challenges in handling small objects. A traditional approach to improve small-object segmentation is to add a P2 feature layer for enhanced feature capture; however, this method significantly increases computational complexity and post-processing time. To address these challenges while maintaining computational efficiency, we propose an optimized method based on the Path Aggregation Feature Pyramid Network (PAFPN), called the Small Object Enhance Pyramid (SOEP). Rather than directly adding a P2 layer, we employ Spatial-Depthwise Separable Convolution (SPD-Conv) [35] to process the P2 layer, extracting richer, small-object-specific features and fusing them with the P3 layer. An illustration of the SPD-Conv process is provided in Figure 5. This fusion strategy effectively preserves key small-object information while mitigating the rise in computational costs. Furthermore, by incorporating the Cross-Stage Partial Network (CSP) [36] architecture and OmniKernel-based [37] feature aggregation (CSP-OmniKernel), we enhance feature representation, as illustrated in Figure 6. The OmniKernel module, composed of global, coarse-scale, and fine-scale branches, efficiently captures multi-scale features from global to local levels. This multi-branch structure substantially improves the segmentation performance of small objects, especially in complex scenes, making it an effective approach for instance segmentation tasks.

## 3. Experiments and Results

### 3.1. Experimental Environment

To ensure experimental consistency, both the training and testing phases of our model were conducted on a computer workstation equipped with a 12th generation Intel Core i9 processor (by Intel Corporation, Santa Clara, CA, USA) and an NVIDIA GeForce RTX 3080Ti laptop GPU (by NVIDIA Corporation, Santa Clara, CA, USA), operating under Windows 11 (by Microsoft Corporation, Redmond, DC, USA). The machine learning framework utilized for model training was Pytorch, version 2.2.2. Furthermore, we utilized CUDA version 11.8, a general-purpose parallel computing platform for GPUs, with NVIDIA’s CUDA Deep Neural Network library (cuDNN) v8.9.7 to optimize computational efficiency. Parameter settings for CabbageNet training were selected to optimize segmentation performance across diverse scenarios, with a learning rate of 0.01, batch size of 16, momentum of 0.937, weight decay of 0.0005, 300 training epochs, and an input image size of 640 × 640 pixels.

### 3.2. Evaluation Metrics

This study employs a comprehensive set of evaluation metrics to systematically assess the performance of the model, including precision, recall, average precision (AP), mean average precision (mAP), Intersection over Union (IoU), Dice Coefficient, F1 Score, Hausdorff Distance, Pixel Accuracy (PA), Params, GFlOPS (giga floating-point operations per second), and FPS (frames per second). Precision, defined by Equation (Equation 1), measures the model’s ability to correctly identify positive samples. Recall, calculated using Equation (Equation 2), represents the proportion of actual positive samples successfully detected by the model. AP, defined by Equation (Equation 3), evaluates detection and classification performance across varying confidence thresholds, while mAP, given by Equation (Equation 4), provides an aggregated measure of precision across multiple categories. IoU, defined by Equation (Equation 5), quantifies spatial overlap between predicted and ground-truth regions as the ratio of their intersection to union. The Dice Coefficient, detailed in Equation (Equation 6), assesses the similarity between predicted and actual regions, with a focus on overlap quality. F1 Score, represented by Equation (Equation 7), harmonizes precision and recall in a single metric. Furthermore, Hausdorff Distance, described in Equation (Equation 8), measures the maximum discrepancy between predicted and ground-truth boundaries. PA, as defined by Equation (Equation 9), represents the ratio of correctly classified pixels to the total number of pixels. Params, as defined by Equation (Equation 10), denotes the total number of learnable parameters in the model. GFlOPS, given in Equation (Equation 11), quantifies the number of floating-point operations per second during inference, while FPS, described in Equation (Equation 12), indicates the number of frames processed per second—a key metric for assessing real-time performance.
(1)$$\;\mathrm{Precision}\;=\frac{TP}{TP+FP}$$
(2)$$\;\mathrm{Recall}\;=\frac{TP}{TP+FN}$$
(3)$$\mathrm{AP}=\int_{0}^{1}\;\mathrm{Precision}\;\left(\;\mathrm{Recall}\;\right)d\left(\;\mathrm{Recall}\;\right)$$
(4)$$\mathrm{mAP}=\frac{1}{N}\sum_{i=1}^{N}AP_{i}$$
(5)$$\mathrm{IoU}=\frac{TP}{TP+FP+FN}$$
(6)$$\mathrm{Dice}=\frac{2\cdot TP}{2\cdot TP+FP+FN}$$
(7)$$F1=2\cdot \frac{\mathrm{Precision}\cdot \mathrm{Recall}}{\mathrm{Precision}+\mathrm{Recall}}$$
(8)$$d_{H}\left(X,Y\right)=\max\left\{\underset{x\in X}{\sup}\underset{y\in Y}{\inf}d\left(x,y\right),\underset{y\in Y}{\sup}\underset{x\in X}{\inf}d\left(x,y\right)\right\}$$
(9)$$\mathrm{PA}=\frac{TP+TN}{TP+TN+FP+FN}$$
(10)$$\mathrm{Params}=\sum_{l=\mathrm{conv}}\left(\left(K_{h}\times K_{w}\times C_{\mathrm{in},l}+1\right)\times C_{\mathrm{out},l}\right)+\sum_{l=\mathrm{fc}}\left(C_{\mathrm{in},l}\times C_{\mathrm{out},l}+C_{\mathrm{out},l}\right)$$
(11)$$\mathrm{GFlOPS}=\frac{\mathrm{Total}\;\mathrm{Floating}\;\mathrm{Point}\;\mathrm{Operations}}{\mathrm{Execution}\;\mathrm{Time}\;\left(\mathrm{seconds}\right)}\times 10^{-9}$$
(12)$$\mathrm{FPS}=\frac{\mathrm{Total}\;\mathrm{Number}\;\mathrm{of}\;\mathrm{Frames}\;\mathrm{Processed}}{\mathrm{Total}\;\mathrm{Time}\;\left(\mathrm{seconds}\right)}$$
where TP denotes true positive, FP denotes false positive, FN denotes false negative, *N* denotes the number of categories, $AP_{i}$ denotes the AP value corresponding to category *i*, sup represents the supremum (least upper bound) and inf represents the infimum (greatest lower bound), while $K_{h}$ and $K_{w}$ represent the height and width of the convolution kernel, $C_{\mathrm{in},l}$ and $C_{\mathrm{out},l}$ denote the input and output channels of the *l*-th layer, respectively, and “+1” denotes the bias term.

### 3.3. Comparison Experiments

To validate the effectiveness of the proposed CabbageNet model in the instance segmentation task for harvested cabbage heads in complex scenes, we conducted a systematic evaluation using COCO metrics, including Mask AP0.5:0.95, Mask AP0.5, Mask AP0.75, Mask APsmall, parameter count, FPS, and GFLOPS [38]. We compared CabbageNet with several state-of-the-art instance segmentation models, including Mask R-CNN, Cascade Mask R-CNN, Mask Scoring R-CNN, Hybrid Task Cascade, YOLACT, SOLO, PointRend, SOLOv2, QueryInst, and ConvNeXt-V2, with the experimental results presented in Table 1.

The analysis reveals that CabbageNet presents superior performance across multiple key metrics, achieving Mask AP0.5:0.95 and Mask AP0.5 values of 78.9% and 94.0%, respectively, significantly surpassing other models such as Mask R-CNN (74.1%, 90.8%) and Cascade Mask R-CNN (74.5%, 90.2%) in the same metrics. Furthermore, CabbageNet excels in Mask AP0.75 and Mask APsmall, reaching 85.4% and 38.7%, respectively, far exceeding YOLACT (73.6% and 31.0%) and SOLO (62.8% and 7.0%). In terms of computational efficiency, CabbageNet’s parameter count is only 3.21 M, with a GFLOPS of 15.1 and an FPS of 154, indicating remarkable computational performance while maintaining high accuracy. This advantage is particularly evident when compared to the larger parameter models, such as Cascade Mask R-CNN (76.8M, FPS of 12) and Hybrid Task Cascade (79.9M, FPS of 6). Additionally, CabbageNet has proven its adaptability in complex scenarios, effectively handling diverse image datasets, particularly excelling in harvested cabbage head instance segmentation. Consequently, the experimental results indicate that CabbageNet demonstrates significant superiority in this specific task.

To evaluate the effectiveness of the proposed model, a comparative analysis was conducted with the advanced segmentation model SAM (Segment Anything Model) and its derivatives, which do not rely on prior knowledge. Since SAM models perform segmentation across all elements within an image, post-processing was necessary to refine their outputs. Specifically, an IoU matrix combined with the Hungarian algorithm was applied to match predicted masks with ground-truth masks, enabling precise identification and extraction of cabbage head targets. To ensure a comprehensive performance assessment, universal metrics were employed to evaluate segmentation accuracy, consistency, and boundary quality. The detailed results of the comparison are summarized in Table 2.

The experimental results clearly demonstrate that CabbageNet outperforms all other models in the SAM series across key metrics. It achieves the highest values in Intersection over Union (IoU), Dice Coefficient, and F1 Score, with 85.3%, 90.0%, and 90.0%, respectively, significantly surpassing SAM2, which shows notably lower performance. CabbageNet also excels in boundary precision, with a Hausdorff Distance of 27.1 pixels, much lower than SAM2 at 175.4 pixels. In Pixel Accuracy (PA), it achieves 99.4%, outperforming MobileSAM at 97.2%. Furthermore, CabbageNet attains a remarkable frame rate of 154 FPS, surpassing FastSAM at 63 FPS. These results underscore CabbageNet’s superior segmentation accuracy, boundary precision, and computational efficiency, making it a highly competitive model for real-time applications that require both high precision and speed.

To further evaluate the effectiveness of the CabbageNet model for segmenting mature cabbage heads in complex environments, comparative experiments were conducted against YOLOv5n-seg, YOLOv9c-seg, and the baseline model YOLOv8n-seg. The results of the experiment are presented in Table 3.

The findings indicate that CabbageNet exhibited superior performance compared to the competing models in both mask precision and mean average precision (mAP). In particular, the CabbageNet model exhibited a mask precision of 92.2%, which was markedly higher than that of the YOLOv8n-seg (90.9%) and YOLOv5n-seg (90.5%) models. This illustrates its enhanced capability in addressing intricate agricultural segmentation tasks. Although YOLOv9c-seg exhibited a marginally higher mAP50 (95.4%), CabbageNet demonstrated a superior balance between accuracy and computational efficiency, with a parameter count of 3.21M and a model size of 6.46MB. The computational load of CabbageNet was 15.1 GFLOPs, which was slightly higher than that of YOLOv8n-seg (12.0 GFLOPs). However, this increase is justified given the significant increase in precision observed. CabbageNet exhibited an inference speed of 154 FPS, which, although inferior to that of YOLOv5n-seg (329 FPS) and YOLOv8n-seg (215 FPS), was considerably higher than that of YOLOv9c-seg (22 FPS), thus meeting the requisite specifications for real-time agricultural applications. Overall, CabbageNet demonstrated substantial improvements in segmentation accuracy, model compactness, computational efficiency, and real-time performance, making it well suited for instance segmentation tasks in automated cabbage head harvesting under visually challenging conditions.

### 3.4. Ablation Experiments

In order to demonstrate the performance improvement contributed by each module, we conducted ablation experiments to observe the impact of gradually incorporating distinct modules on model performance. In these experiments, the C2f-DAttention module, the ADown module, and the SOEP module were sequentially added to evaluate the specific effects of adding modules on the overall model performance. The results of these experiments are presented in Table 4.

The results of the ablation experiments with various modules indicate that the proposed CabbageNet model demonstrates significant advantages in terms of accuracy and mAP. Compared to the benchmark model YOLOv8n-seg, CabbageNet shows an improvement in precision from 90.9% to 92.2%, while maintaining a stable mAP50 of 95.1% and achieving a high recall rate of 87.2% through the integration of modules such as C2f-DAttention, ADown, and SOEP. The dynamic attention mechanism within C2f-DAttention enhances responsiveness to key features and effectively reduces background noise, significantly improving the model’s overall detection and segmentation accuracy in challenging environments. However, this focus on prominent features slightly deprioritizes the importance of secondary features, contributing to a marginal decrease in recall. The incorporation of the ADown module not only reduces the number of model parameters in an efficient manner but also preserves the integrity of the features through a rational downsampling approach. This strategy achieves the objective of optimizing the model complexity and enhancing the detection efficiency. The incorporation of the ADown module not only effectively reduces the number of parameters in the model, but also maintains the integrity of the features through a reasonable downsampling strategy. Additionally, the introduction of SOEP enhances the aggregation of features across different scales, further improving precision.

This strategy achieves a balance between optimizing the complexity of the model and improving the detection efficiency. Although these improvements do increase the computational burden of the model to a certain extent, resulting in a reduction in the inference speed, it remains capable of achieving the real-time segmentation of cabbage heads during the harvesting period. In conclusion, the improvement strategy of CabbageNet, through multi-module co-optimization, makes the model achieve significant improvement in the core metrics such as precision and mAP, which proves the effectiveness of our proposed method in the segmentation task.

### 3.5. Visualization and Analysis of Experiments

To further verify the segmentation effect of the proposed model and the comparison model more intuitively, we compared the same image after segmentation. The segmentation effect is shown in Figure 7, where the first is the original image and the rest are the segmentation mask maps of other mainstream models and the CabbageNet model, and the corresponding segmentation model is shown under each image.

In evaluating the performance of various instance segmentation algorithms on the same cabbage image, it was found that the segmentation masks generated by Cascade Mask R-CNN, ConvNeXt-V2, Hybrid Task Cascade, Mask R-CNN, Mask Scoring R-CNN, QueryInst, SOLOv2, and YOLACT did not fully cover the cabbage head. Although PointRend showed relatively better performance, it still exhibited deficiencies in edge segmentation. Moreover, none of these models succeeded in segmenting the small, incomplete cabbage head in the upper right corner. YOLOv5n-seg failed to completely segment the larger cabbage head, while YOLOv8n-seg incorrectly split the larger cabbage head into two overlapping instances, creating a bluish-purple overlap. YOLOV9c-seg provided better edge handling but showed slight inaccuracies in segmenting the medium-sized cabbage. In contrast, CabbageNet handles edge details with precision, outperforming all other models in overall segmentation quality.

As shown in Figure 8, this study presents a comparative analysis of the segmentation performance between the SAM series models and the proposed CabbageNet. The results reveal that SAM series models often struggle to differentiate objects that closely resemble the background, leading to significant segmentation errors and producing segmentation edges that lack precision. In contrast, CabbageNet effectively addresses these challenges by minimizing such accuracy-reducing errors. CabbageNet demonstrates superior performance, accurately and efficiently segmenting cabbage heads in complex cabbage harvesting environments.

To further validate the segmentation performance of the model in real-world production environments, an image acquisition device was installed on the harvesting equipment, leaf-stripping device, and high-altitude UAV of the cabbage harvesting machinery. The systems captured images from multiple perspectives, thereby reflecting the complex conditions typically encountered in agricultural settings. The collected images were used to compare the segmentation of cabbage heads in complex backgrounds, using both the YOLO series segmentation models and the proposed CabbageNet model. The detailed segmentation results are shown in Figure 9.

As shown in the results, among the images from the harvesting device, the YOLOv5n-seg model performed poorly in detecting and segmenting small-sized targets. In contrast, CabbageNet consistently outperformed the other models, demonstrating superior performance in both detection and segmentation. Among the images from the leaf-stripping device, YOLOv5n-seg and YOLOv9c-seg both misidentified the stripped cabbage leaves as cabbage heads, while YOLOv8n-seg mistakenly included the cabbage leaves in its segmentation mask. However, CabbageNet provided significantly more accurate segmentation, particularly in distinguishing cabbage heads from surrounding leaves. Similarly, in the images acquired by the UAV, CabbageNet outperformed the other models in segmenting small cabbage heads. Overall, CabbageNet consistently delivered superior segmentation performance across complex backgrounds, performing better than the other YOLO series models.

The ablation experiment results are visually compared and presented in Figure 10. An analysis of the figure reveals that the YOLOv8n-seg model exhibits slightly lower segmentation accuracy for both single targets and multiple small-sized targets. However, after incorporating the C2f-DAttention module, a significant improvement in accuracy is observed. Further integration of the ADown module not only reduces the number of parameters and computational complexity considerably but also preserves the model’s performance. Finally, the addition of the SOEP module leads to a substantial enhancement in segmentation performance for small-sized targets. Overall, the various improvements proposed in this study contribute positively to the model’s performance in segmentation tasks.

## 4. Conclusions and Discussion

This paper addresses the challenges of cabbage detection and segmentation in unmanned cabbage harvesting robots operating in complex environments. To reduce the miss rate and damage rate during automated cabbage harvesting, an improved cabbage head segmentation algorithm, CabbageNet, is proposed based on YOLOv8n-seg. First, a dynamic attention mechanism is introduced into the C2f module to create the C2f DAttention module, enhancing segmentation accuracy for cabbages of varying sizes. Next, the downsampling mechanism is improved with the ADown module, which significantly reduces the number of parameters and computational complexity while maintaining high-level feature extraction capabilities. Finally, the SOAP module is integrated to boost the recognition and segmentation of small-sized cabbages. Experimental results show that the CabbageNet model, with a size of only 6.46MB, achieves a Mask Precision of 92.2%, a Mask Recall of 87.2%, and a Mask mAP50 of 95.1%, with a segmentation speed of 154 FPS in real-time applications. These results provide a solid foundation for the application of unmanned cabbage harvesting robots in complex harvesting environments.

The dataset used in this study includes only a limited number of cabbage varieties. Consequently, the model trained on this dataset may face difficulties in generalizing to new or unseen varieties not included in the dataset. To enhance the model’s robustness and extend its applicability, further expansion of the dataset will be required to accommodate a broader range of cabbage types. In the future, the current dataset will be expanded by the addition of further cabbage varieties, thereby enhancing the model’s capacity to recognize and segment multiple types of cabbage. Furthermore, the CabbageNet algorithm will be optimized to enable its effective operation in unmanned cabbage harvesting environments, which are characterized by complex and variable conditions.

## Figures and Tables

**Figure 1 sensors-24-08115-f001:**
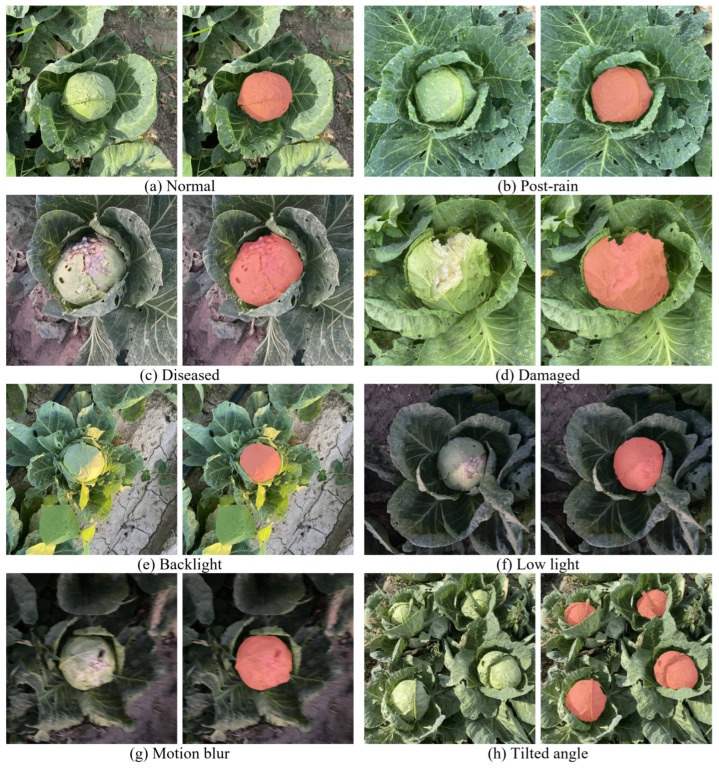
Comparison of cabbage before and after labeling. Each pair of images is shown together, with the original on the left and the labeled version on the right.

**Figure 2 sensors-24-08115-f002:**
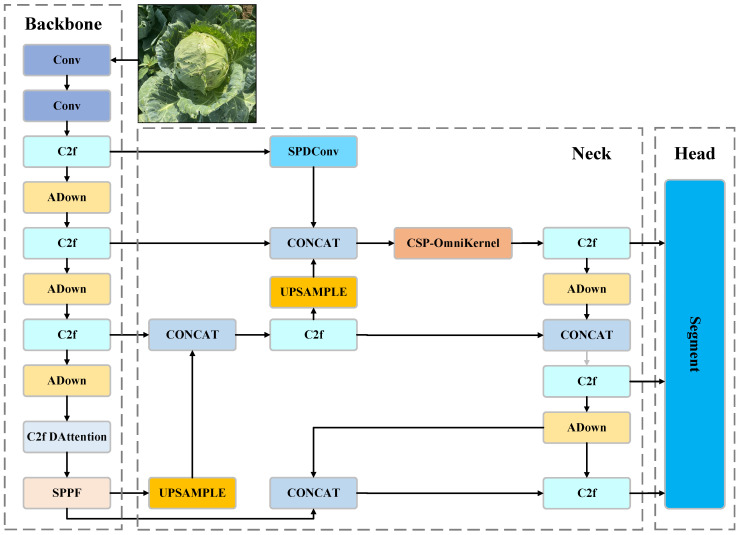
CabbageNet model structure.

**Figure 3 sensors-24-08115-f003:**
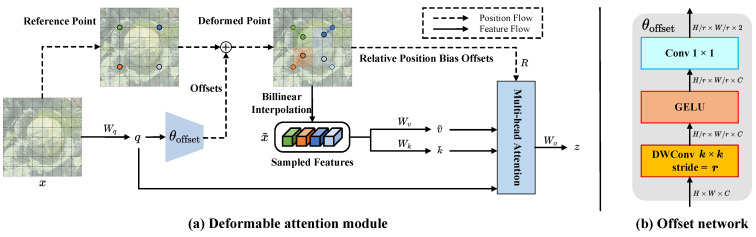
An Illustration of the deformable attention mechanism. (**a**) Information flow of the deformable attention. (**b**) Structure of the offset generation network. DWConv represents depthwise convolution.

**Figure 4 sensors-24-08115-f004:**
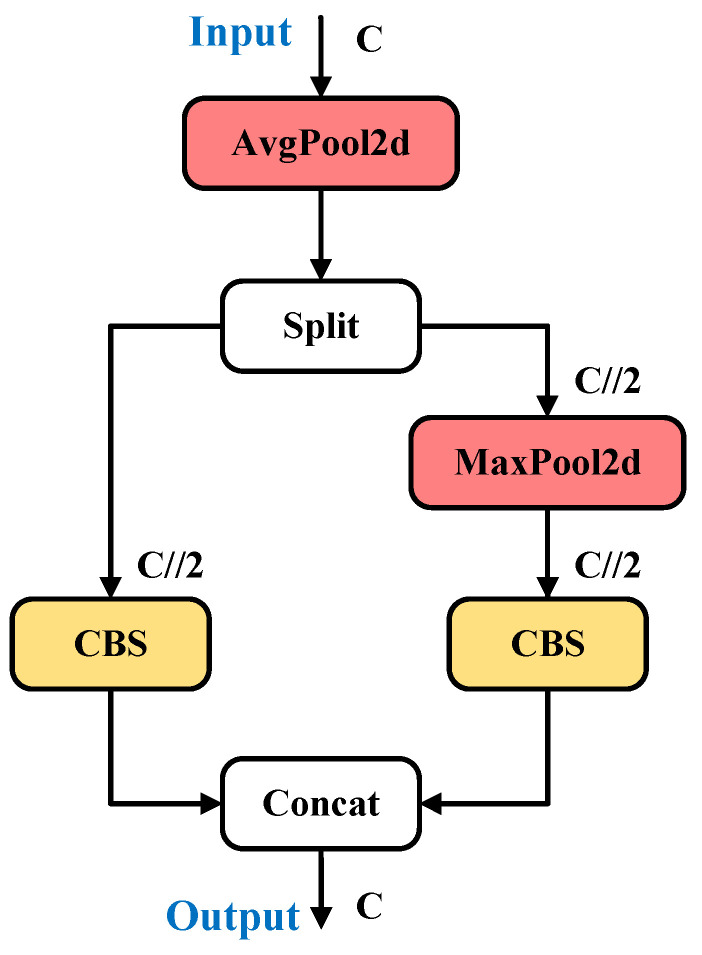
ADown module network diagram. CBS denotes a sequence of operations comprising convolution, batch normalization, and SiLU activation. AvgPool2d and MaxPool2d denote average pooling and max pooling layers, respectively.

**Figure 5 sensors-24-08115-f005:**
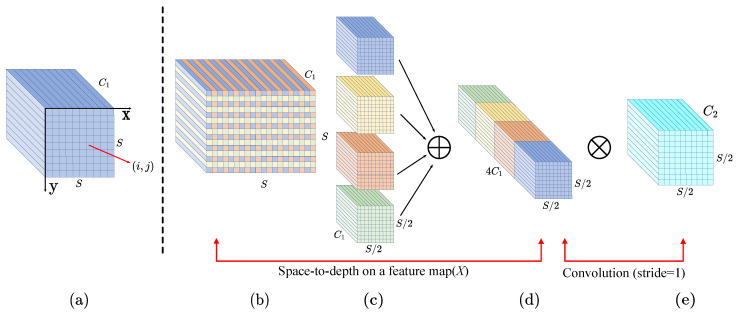
An Illustration of the SPD-Conv Process. (**a**) Traditional feature map. (**b**) Space-to-depth transformation. (**c**) Channel merging. (**d**) Addition operation. (**e**) Convolution with no stride.

**Figure 6 sensors-24-08115-f006:**
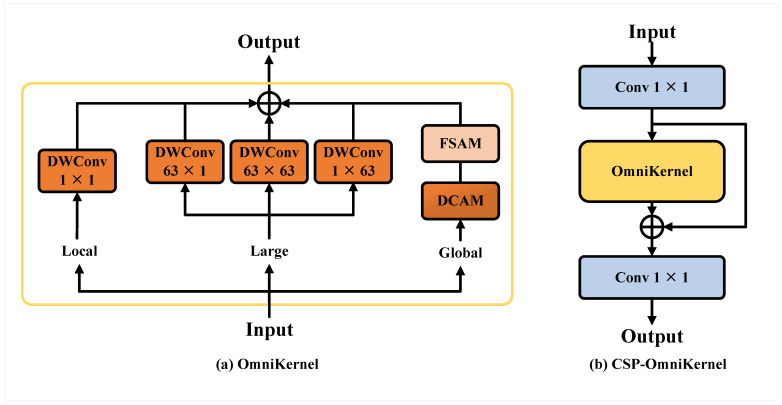
An illustration of the OmniKernel and CSP-OmniKernel. (**a**) Structure of the OmniKernel module, showing local, large, and global branches. (**b**) CSP-OmniKernel architecture, incorporating the OmniKernel module for enhanced feature aggregation. DWConv represents depthwise convolution, DCAM represents Dual-Domain Channel Attention Module, and FSAM represents Frequency-Based Spatial Attention Module.

**Figure 7 sensors-24-08115-f007:**
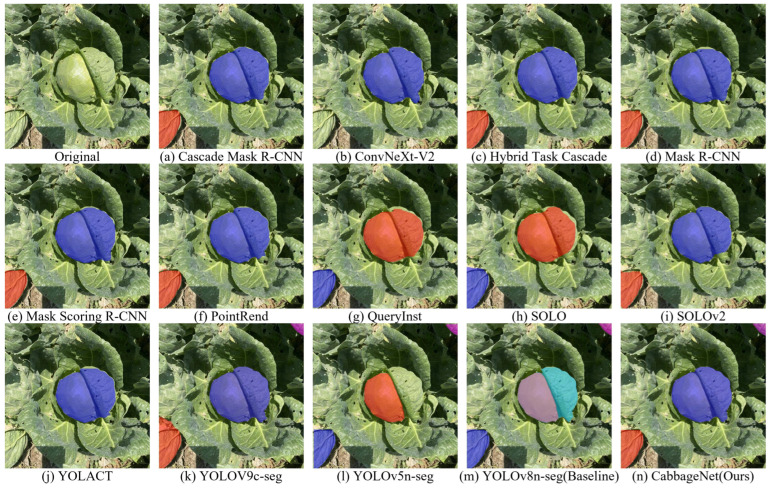
Comparison of segmentation results from different instance segmentation models. The color sequence of segmented instances in the figure is as follows: blue, orange-yellow, purple, indigo, etc.

**Figure 8 sensors-24-08115-f008:**
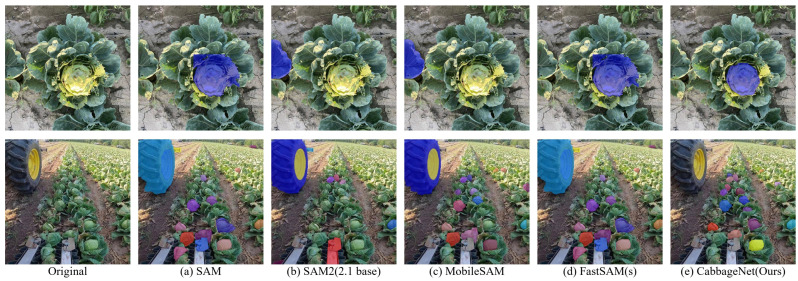
Comparison of segmentation results from SAM series models. The color sequence of segmented instances in the figure is as follows: blue, orange-yellow, purple, indigo, etc.

**Figure 9 sensors-24-08115-f009:**
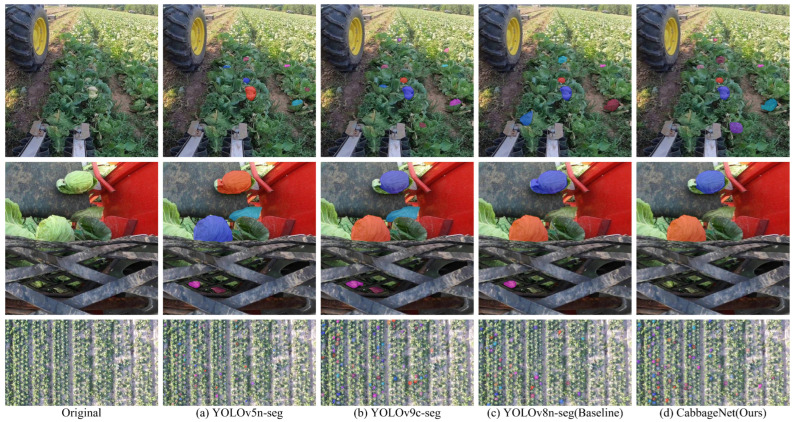
Comparison of segmentation performance between YOLO series models and CabbageNet in a real-world cabbage harvesting environment. The color sequence of segmented instances in the figure is as follows: blue, orange-yellow, purple, indigo, etc.

**Figure 10 sensors-24-08115-f010:**
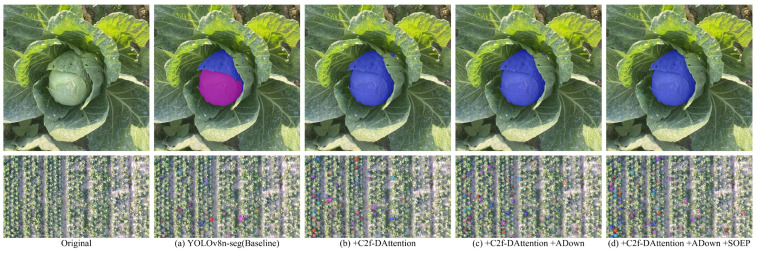
Visualization comparison of ablation experiments. The color sequence of segmented instances in the figure is as follows: blue, orange-yellow, purple, indigo, etc.

**Table 1 sensors-24-08115-t001:** Comparison experiments with classic models.

Models	Mask AP0.5:0.95/%	Mask AP0.5/%	Mask AP0.75/%	Mask APsmall/%	Params/M	GFLOPS	FPS
Mask R-CNN	74.1	90.8	80.2	36.3	43.9	284.6	21
Cascade Mask R-CNN [39]	74.5	90.2	80.4	35.5	76.8	3323.2	12
Mask Scoring R-CNN [40]	75.2	89.8	80.5	35.3	60.4	366.6	23
Hybrid Task Cascade [41]	75.9	91.7	81.8	38.0	79.9	3506.0	6
YOLACT	68.2	89.4	73.6	31.0	34.7	163.2	21
SOLO [42]	58.2	78.1	62.8	7.0	35.9	311.5	11
PointRend [43]	73.3	90.5	79.6	35.1	55.9	184.1	18
SOLOv2 [44]	49.3	76.1	50.7	7.8	46.0	276.7	12
QueryInst [45]	68.4	85.0	74.6	24.4	172.2	135.6	10
ConvNeXt-V2	75.8	89.0	80.5	34.3	108.1	469.6	6
CabbageNet (Ours)	78.9	94.0	85.4	38.7	3.21	15.1	154

**Table 2 sensors-24-08115-t002:** Comparison experiments with the SAM series models.

Models	IoU/%	Dice/%	F1/%	Hausdorff Distance/Pixels	PA/%	FPS
SAM (base) [46]	70.2	74.2	74.1	73.8	96.8	-
SAM2 (2.1 base) [47]	49.9	51.9	51.9	175.4	94.9	-
MobileSAM [48]	76.1	80.3	80.3	58.45	97.2	-
FastSAM (s) [49]	71.3	76.6	76.6	80.3	96.9	63
CabbageNet (Ours)	85.3	90.0	90.0	27.1	99.4	154

**Table 3 sensors-24-08115-t003:** Comparison experiments with the YOLO series models.

Models	Mask Precision/%	Mask Recall/%	Mask mAP50/%	Mask mAP50-95/%	Params/M	FPS	GFLOPs	Model Size/MB
YOLOv5n-seg [50]	90.5	88.5	94.6	76.5	1.88	329	6.7	3.96
YOLOv9c-seg	90.6	89.5	95.4	82.5	27.63	22	157.6	56.30
YOLOv8n-seg (Baseline)	90.9	88.0	94.9	80.4	3.26	215	12.0	6.50
CabbageNet (Ours)	92.2	87.2	95.1	80.6	3.21	154	15.1	6.46

**Table 4 sensors-24-08115-t004:** Results of the various ablation experiments.

Models	C2f-DAttention	ADown	SOEP	Mask Precision/%	Mask Recall/%	Mask mAP50/%	Mask mAP50-95/%	Params/M	FPS	GFLOPs	Size/MB
YOLOv8n-seg (Baseline)	–	–	–	90.9	88.0	94.9	80.4	3.26	215	12.0	6.50
+ C2f-DAttention	✓	–	–	92.0	86.7	94.9	80.5	3.32	193	12.0	6.64
+ C2f-DAttention + ADown	✓	✓	–	91.6	87.2	95.1	80.6	2.91	162	11.3	5.87
CabbageNet (Ours)	✓	✓	✓	92.2	87.2	95.1	80.6	3.21	154	15.1	6.46

## Data Availability

The data supporting the findings of this study are available upon request from the readers.

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
