# Peer review of "CabbageNet: Deep Learning for High-Precision Cabbage Segmentation in Complex Settings for Autonomous Harvesting Robotics"

_sensors, 2024, doi:10.3390/s24248115_

Round 1

Reviewer 1 Report

Comments and Suggestions for Authors

The author conducted research on the image recognition and segmentation of cabbages and made four contributions: (1) Construction of a cabbage instance segmentation dataset. (2) Integration of deformable attention into the C2f module. (3) Improvement of the downsampling process using the ADown module. (4) Enhancement of small object segmentation using the SOEP module. It has a certain degree of novelty and innovativeness. However, it should be improved from the following aspects.

1. In the third paragraph of the introduction section, it is a review of using deep learning for image segmentation. However, it seems to be just a list of previous works and has little relevance to your work. You should review more image segmentation techniques for the problem of "complex background" and lead to the necessity of this paper's work.

2. There seems to be a problem with the organization of your paper: Section 2.3 is titled "Experimental Methods", and Section 3 is "Experiments and Results", and there seems to be a semantic overlap between them. Usually, in Section 2, you can introduce the improved model, and in Section 3, introduce your experiments, including experimental settings, evaluation criteria, etc.

3. Your model has made three improvements. But there is a lack of certain analysis on why these three improvements are better than the original model YOLOv8n. 

4. The comparison experiment did not compare the accuracy with the baseline, that is, YOLOv8n.

5. From your ablation experiment, the accuracy improvement of your model is obvious, and the recall rate has even decreased.

Author Response

Comments 1: In the third paragraph of the introduction section, it is a review of using deep learning for image segmentation. However, it seems to be just a list of previous works and has little relevance to your work. You should review more image segmentation techniques for the problem of "complex background" and lead to the necessity of this paper's work.
Response 1: Thank you for pointing this out. We agree with this comment. Therefore, we have revised the third paragraph of the introduction section you mentioned, emphasizing “complex background” and adding additional references and relevant content related to “complex background”.

Comments 2: There seems to be a problem with the organization of your paper: Section 2.3 is titled "Experimental Methods", and Section 3 is "Experiments and Results", and there seems to be a semantic overlap between them. Usually, in Section 2, you can introduce the improved model, and in Section 3, introduce your experiments, including experimental settings, evaluation criteria, etc.
Response 2: Thanks to your reminder, I rechecked my manuscript and there was indeed such a problem, so I have organized some of the paragraphs in Section 2 and Section 3 to be more in line with the canonical essay structure.

Comments 3: Your model has made three improvements. But there is a lack of certain analysis on why these three improvements are better than the original model YOLOv8n. 
Response 3: Thank you for pointing this out. We agree with this comment. Therefore, we have added visualization and analysis of the ablation experiments associated with the Improvement section, see the Visualization and analysis of experiments section of the manuscript, and we have highlighted the additions. 

Comments 4: The comparison experiment did not compare the accuracy with the baseline, that is, YOLOv8n.
Response 4: Thank you for the reminder. Essentially, I added experiments with the baseline model YOLOv8n-seg to my comparison with the YOLO series models, but I neglected to note this in the Table 3. I have since revised my manuscript and corrected the clerical errors.

Comments 5: From your ablation experiment, the accuracy improvement of your model is obvious, and the recall rate has even decreased.
Response 5: Yes, the dynamic attention mechanism in the improved C2f-DAttention in our proposed model enhances the responsiveness to critical features and effectively reduces the background noise to improve the accuracy, but this attention to salient features slightly reduces the importance of secondary features, resulting in a slight decrease in recall. We describe the relevant content in the manuscript [lines 336-340] and revise the penmanship section.

Reviewer 2 Report

Comments and Suggestions for Authors

The study proposes an interesting deep learning method (YOLOv8n-seg) for cabbage segmentation in complex agricultural environments, which is an important contribution to the field of autonomous harvesting. The authors present some novel ideas, such as the use of deformable attention and dynamic sampling, which appear promising. However, there are several areas where the paper can be improved:

1.What environmental conditions were the dataset samples in Figure 1 collected under? Can this be specified and labeled in detail?

2.In Section 3.2, only the evaluation metrics for the ablation experiments are provided. More specific visual prediction results should be given.

3.In Section 2.3, the description of the method proposed by the authors is not prominent enough; most of it is an introduction and description of existing work.

4.In the quantitative results, the meanings of some parameters are not specifically explained, such as Params, GFlOPS, FPS.

5.What are the main limitations of this study, and how can future research overcome these limitations? These should be specifically provided in the conclusion section.

Author Response

Comments 1: What environmental conditions were the dataset samples in Figure 1 collected under? Can this be specified and labeled in detail?
Response 1: Thank you for your suggestion, which I agree with, so I have modified Figure 1 with a note about the environmental conditions under which the dataset samples were collected.

Comments 2: In Section 3.2, only the evaluation metrics for the ablation experiments are provided. More specific visual prediction results should be given.
Response 2: Thank you for pointing this out. We agree with this comment. Therefore, we have added visualization and analysis of the ablation experiments associated with the Improvement section, see the Visualization and analysis of experiments section of the manuscript, and we have highlighted the additions. 

Comments 3: In Section 2.3, the description of the method proposed by the authors is not prominent enough; most of it is an introduction and description of existing work.
Response 3:Thank you for pointing this out, we agree and have made changes to section 2.3, specifically in lines 156-172.

Comments 4: In the quantitative results, the meanings of some parameters are not specifically explained, such as Params, GFlOPS, FPS.
Response 4:Thanks to your suggestion, I have explained the metrics Params, GFlOPS, and FPS in section 3.2 and provided the corresponding formulas, with the changes in lines 250-255.

Comments 5: What are the main limitations of this study, and how can future research overcome these limitations? These should be specifically provided in the conclusion section.
Response 5:Thanks to your reminder, I have added a "limitations" statement to the conclusion of the paper, in lines 428-432 of the manuscript.

Round 2

Reviewer 1 Report

Comments and Suggestions for Authors

The author still needs to revise the review section, as it lacks an analysis of the advantages and disadvantages of existing methods, as well as an explanation of the necessity of conducting the current research.

For example, Dandan Wang et al. [18]  achieved accuracies of 93.2% for bounding box detection and 92.0% for instance segmentation in complex backgrounds. The author’s model achieved precise of 92.2%, which is not higher than the previous study. Why is it necessary to conduct the current research?

The author just state "However, accurate and efficient segmentation becomes challenging for fruits and vegetables moving within complex backgrounds. "  Since your work is not more accurate than others', does your model perform more effectively in terms of lightweight performance and mobility for fruits and vegetables? Additionally, how fast can the harvester move? Does it affect the recognition accuracy?

Author Response

Comments 1: The author still needs to revise the review section, as it lacks an analysis of the advantages and disadvantages of existing methods, as well as an explanation of the necessity of conducting the current research.
Response 1: Thank you for your suggestion. We agree with this comment. Therefore, I have rechecked the entire review section, carefully examining the literature and providing an analysis of the strengths and weaknesses of the existing methods, and have rewritten the section accordingly. Additionally, I have revised the summary paragraph to ensure consistency with the rest of the review. Furthermore, regarding cabbage harvesting machinery in unmanned harvesting scenarios, it typically operates at speeds of 0.6-0.8 km/h,which has some real-time requirements for segmentation.